# A Practical Framework for ASFV Disinfectant Evaluation: Differentiating Cytopathic Effects from Cytotoxicity via Integrated Analytical Methods

**DOI:** 10.3390/pathogens14050451

**Published:** 2025-05-04

**Authors:** Sok Song, Kyu-Sik Shin, Su-Jeong Kim, Yong Yi Joo, Bokhee Han, So-Hee Park, Hyun-Ok Ku, Wooseog Jeong, Choi-Kyu Park

**Affiliations:** 1Veterinary Drugs & Biologics Division, Animal and Plant Quarantine Agency, Gimcheon 39660, Gyeongbuk do, Republic of Korea; ssoboro@korea.kr (S.S.); soso9354@naver.com (K.-S.S.); kimsujeong27@korea.kr (S.-J.K.); wndyddl7091@korea.kr (Y.Y.J.); joy2996@korea.kr (B.H.); sh0526@korea.kr (S.-H.P.); kuho@korea.kr (H.-O.K.); 2College of Veterinary Medicine & Animal Disease Intervention Center, Kyungpook National University, Daegu 41566, Gyeongbuk do, Republic of Korea

**Keywords:** ASFV, disinfection, virucidal efficacy testing, cytotoxicity, CPE

## Abstract

African swine fever virus (ASFV) is a highly virulent DNA virus that has spread globally since its introduction into Georgia in 2007, causing substantial economic losses in the swine industry. In the absence of an effective vaccine, chemical disinfection remains a key strategy for disease control. However, in cell-based disinfectant efficacy testing, distinguishing between disinfectant-induced cytotoxicity and virus-induced cytopathic effects (CPEs) remains a major challenge, leading to the potential misinterpretation of results. To address this, we developed a multi-step analytical framework to differentiate CPEs from cytotoxicity using a Vero cell-adapted ASFV strain. Virkon^®^ S was tested at three dilutions—375×, 275× (manufacturer-recommended), and 175×—and evaluated through CPE observation, lactate dehydrogenase (LDH) and 3-(4,5-dimethylthiazol-2-yl)-2,5-diphenyltetrazolium bromide (MTT) assays, and antigen detection via lateral flow immunoassay (p30) and immunofluorescence (p54). Notably, the 375× dilution achieved effective viral inactivation with significantly lower cytotoxicity, demonstrating that this framework can facilitate a more refined determination of disinfectant working dilutions. Furthermore, increased p30 signals after disinfection and the observation of lower cytotoxicity in virus-plus-disinfectant groups compared to disinfectant-only groups highlight the complexity of virus-disinfectant interactions and the potential for misinterpretation. This study provides a standardized and interpretable strategy for assessing ASFV disinfectant efficacy and offers a practical basis for evaluating other enveloped viruses in future disinfection studies.

## 1. Introduction

African swine fever virus (ASFV) is the causative agent of African swine fever (ASF), a highly lethal disease affecting domestic pigs and wild boars. First reported in Kenya in the 1920s, ASFV remained endemic in sub-Saharan Africa for decades. A major turning point occurred in 2007, when ASFV was introduced into Georgia, likely through contaminated food waste. Since then, the virus has spread across Eastern and Central Europe and Russia and reached China in 2018, triggering an unprecedented outbreak across Asia. ASFV has since been reported in Vietnam, Republic of Korea, the Philippines, Thailand, and other countries, causing a severe disruption in the global swine industry [1,2]. This rapid and extensive spread, combined with the lack of an effective vaccine and the virus’s environmental resilience, has made ASF one of the most challenging transboundary animal diseases to control [3,4]. With mortality rates approaching 100% in susceptible populations, ASF has caused significant economic losses worldwide [5,6,7]. ASFV, a large, enveloped, double-stranded DNA virus of the *Asfarviridae* family, possesses a genome ranging from 170 to 193 kbp and encodes over 150 proteins involved in viral replication, immune evasion, and host–cell interactions [8,9,10]. Notably, despite being a DNA virus, ASFV utilizes both the nucleus and cytoplasm of infected cells during its replication cycle, indicating the involvement of complex molecular mechanisms in viral DNA replication, transcription, and virion assembly [6,11]. Due to its genetic and structural complexity, ASFV effectively evades host immune defenses, making it particularly challenging to control. To date, no effective vaccine or antiviral treatment is available; therefore, biosecurity measures, including chemical disinfection, play a crucial role in preventing virus transmission [12,13,14]. The Animal and Plant Quarantine Agency (APQA, Republic of Korea) conducts annual efficacy tests on commercial disinfectants against key pathogens in national animal disease control policies, including ASFV, foot-and-mouth disease virus (FMDV), avian influenza virus (AIV), and porcine epidemic diarrhea virus (PEDV). In Republic of Korea, disinfectant efficacy testing is mandatory not only for the routine surveillance of marketed products but also as a regulatory requirement for product registration. To be approved, disinfectants must demonstrate a ≥10^4^ TCID_50_ reduction in viral titer compared to untreated controls. AIV efficacy is evaluated using the embryonated egg inoculation method, while other viruses are tested via cell-based assays with CPE monitoring. The evaluations involve various disinfectant formulations (e.g., potassium peroxymonosulfate, quaternary ammonium compounds, glutaraldehyde, citric acid) and host cell lines (e.g., Vero, BHK-21, LFBK, MDCK) at manufacturer-recommended dilutions. However, regardless of the chemical composition, virus type, or cell line used, cytotoxicity remains a critical challenge—particularly in tests requiring a 10^4^-fold titer reduction—making it essential to differentiate between disinfectant-induced cytotoxicity and virus-induced CPE.

Cellular damage observed in vitro may arise from two distinct sources: (i) direct cytotoxic effects of disinfectants on cell viability and (ii) virus-induced CPE resulting from viral replication and host cell destruction. Misinterpretation can lead to false conclusions, where cytotoxicity mimics viral CPE (false-positive) or, conversely, an effective disinfectant is underestimated due to a mild cytotoxic effect (false-negative). Given the high cytotoxicity of many disinfectants, a systematic differentiation method is required to ensure reliable ASFV disinfectant evaluation. Previous studies have reported disinfectant-induced cytotoxicity, but most have focused on its presence [15,16] or methods to mitigate it [12,17,18] rather than developing systematic approaches to distinguish it from virus-induced effects. As a result, the lack of standardized analytical approaches has led to potential misinterpretations of virucidal efficacy.

To address this issue, we aimed to develop a robust analytical framework to distinguish disinfectant-induced cytotoxicity from virus-induced CPE. Regulatory guidelines for disinfectant efficacy differ by country. For instance, Republic of Korea and Japan evaluate products based on final formulations, not individual active ingredients [19,20]. This highlights the practical need for a model based on a single product. Therefore, we selected Virkon^®^ S as the model disinfectant based on its widespread international use, its established efficacy against various enveloped viruses including ASFV, and its regulatory recognition, including a unique EPA registration for its specific formulation [21,22,23,24,25]. Although other commercial disinfectants are available, our aim was to first establish a robust analytical framework using a single representative product. The tested dilutions (375×, 275×, and 175×) were selected to bracket the manufacturer-recommended dilution (275×), allowing an evaluation of efficacy at both sub-optimal and potentially excessive cytotoxic concentrations.

A Vero cell-adapted ASFV strain was used as the experimental model for disinfectant efficacy testing. While ASFV naturally targets porcine alveolar macrophages (PAMs), their low reproducibility, limited scalability, and complex culture requirements pose challenges for in vitro studies. Consequently, Vero cells have been widely utilized for ASFV research, including virus isolation, antiviral screening, and disinfectant efficacy testing [6,26]. Vero-adapted ASFV strains, such as BA71V, maintain high replication competence, stable cell propagation, and clear CPE, making them a practical alternative for large-scale screening [27]. This system enables the efficient evaluation of disinfectant efficacy based on CPE analysis. However, some disinfectants exhibited overlapping cytotoxic and virus-induced CPEs, emphasizing the need for a refined analytical approach to accurately distinguish these effects.

ASFV replication proceeds through distinct stages, characterized by early protein expression (e.g., p30) [28,29] and late-stage proteins (e.g., p54) [30,31] involved in virion assembly and release. These viral proteins serve as key markers for evaluating whether disinfectants inhibit viral entry, replication, or assembly. To comprehensively assess disinfectant efficacy, a multi-step analytical approach was employed in this study.

For the quantification of extracellular viral antigens, the AG Rapid Kit was used to detect p30, providing insights into early viral replication and antigen release inhibition by disinfectants [32]. Additionally, an FITC-based immunofluorescence detection of p54 enabled the visualization of late-stage viral protein expression, allowing an evaluation of whether disinfectants effectively block viral replication and virion assembly [33,34]. Cytotoxicity was assessed using the LDH release assay, which quantifies lactate dehydrogenase (LDH) release as an indicator of cell membrane damage caused by disinfectants and viral infection [35]. Moreover, MTT assays were performed to measure mitochondrial activity, offering complementary data on overall cell viability under disinfectant and viral exposure conditions [36,37].

This study aims to systematically analyze disinfectant–virus interactions in cell-based efficacy assays, accurately characterizing time-dependent effects. By moving beyond traditional endpoint observations, this study integrates multiple complementary analytical approaches to enhance experimental interpretability. Ultimately, we propose a robust and reproducible analytical framework that enables precise differentiation between disinfectant-induced cytotoxicity and virus-induced CPE. The proposed approach—integrating antigen detection, immunofluorescence imaging, and cell viability assays—has the potential to contribute to the standardization of disinfectant efficacy evaluations not only for ASFV but also for other enveloped viruses.

## 2. Materials and Methods

### 2.1. Cells and Virus

For all experiments, we exclusively used the ASFV-PJ-VaCIn1 strain (KVCC-VR2400015), which was obtained from the Korea Veterinary Culture Collection (KVCC). This Vero cell-adapted strain—similar to BA71V (accession No. NC_001659)—generates high viral titers and induces clear cytopathic effects (CPEs), enabling reliable and reproducible in vitro assessments. Vero cells (ATCC^®^ CCL-81™, Manassas, VA, USA) were maintained in DMEM (Corning Inc., Corning^®^, New Yrok, NY, USA) supplemented with 10% FBS, Antibiotic-Antimycotic, and 200 mM L-glutamine (Gibco, Thermo Fisher Scientific, Waltham, MA, USA) at 37 °C with 5% CO_2_. Viral stocks were generated by infecting Vero cell monolayers, followed by supernatant collection and clarification via centrifugation at 3000× *g* for 10 min. Viral titers were measured in triplicate using the TCID_50_ method and calculated by the Spearman–Kärber formula. The average titer was 1.3 × 10⁷ TCID_50_/mL (±0.2 × 10⁷), ensuring sufficient viral load for efficacy testing.

### 2.2. Preparation of Disinfectant Solutions

Virkon^®^ S (LANXESS Deutschland GmbH, Cologne, Germany) was used as the disinfectant in this study. To evaluate its efficacy under various conditions, three dilutions (375×, 275×, and 175×) were tested, covering a 100-fold range around the manufacturer-recommended concentration (275×). The dilution conditions were optimized to maintain physiologically acceptable pH and osmolality, minimizing unintended cytotoxic effects.

### 2.3. Experimental Design

We established three experimental groups to evaluate the virucidal efficacy and cytotoxicity of the disinfectant under various conditions. The ASFV + disinfectant (AD) group was prepared by mixing virus suspension with three different dilutions of Virkon^®^ S (375×, 275×, and 175×) in hard water (CureBio, Seoul, Repblic of Korea). The disinfectant-only (D) group consisted of the same Virkon^®^ S dilutions prepared in hard water without the virus. The ASFV-only (A) group, serving as a virus control, used virus suspension mixed with hard water instead of a disinfectant. As illustrated in Figure 1, the inoculum-0 prepared for each group was divided into three parts. First, a portion was immediately tested for viral antigen using a p30 rapid antigen kit. Second, another portion was directly inoculated onto Vero cells, simulating the standard APQA disinfectant efficacy testing procedure. Third, the remaining inoculum was incubated at 37 °C in a CO_2_ incubator for 6 days and then inoculated onto Vero cells to evaluate residual infectivity and delayed cytotoxic effects. All inoculum-0 samples were pre-treated with a neutralizer prior to use. Microscopic observation and LDH assays were conducted daily to monitor cytopathic effects and cytotoxicity. On day 6, cell viability was assessed via an MTT assay. Viral replication was further evaluated through p30 antigen detection (days 0, 6, and 12) and a p54 immunofluorescence assay (day 6). This multi-track approach enabled precise differentiation between disinfectant-induced cytotoxicity and virus-induced CPE under time-dependent conditions.

Three groups were tested: Group A (ASFV only), Group AD (ASFV + disinfectant), and Group D (disinfectant only). Inoculum-0 from each group was divided into three parts: (1) for immediate p30 antigen detection, (2) for direct cell inoculation, and (3) for 6-day incubation at 37 °C prior to inoculation. All inocula were prepared with a neutralizer. Viral antigen (p30) and cytotoxicity (LDH, MTT) were assessed at defined time points, and viral replication was evaluated by p54 immunofluorescence.

### 2.4. Virucidal Efficacy Test and Microscopic Observation

The virucidal efficacy test was conducted using the suspension method following the APQA guidelines. Disinfectants and virus suspensions were mixed at a 1:1 ratio for each dilution and incubated at 4 °C for 30 min, with vortexing every 10 min to ensure a uniform reaction. After incubation, the mixtures were neutralized by adding DMEM supplemented with 10% FBS at a 1:1 ratio. The neutralized solutions were then inoculated into 96-well plates containing Vero cell monolayers at 100 µL per well. Plates were incubated at 37 °C with 5% CO_2_ for six days, and CPEs were monitored daily under a microscope. A scoring system was employed to quantify cell damage severity. TCID_50_ was not included as this study aimed to distinguish cytotoxicity from virus-induced effects through cell-based analyses rather than to quantify virucidal potency. Images of representative wells were captured at each time point to document the progression of cell damages.

### 2.5. Viral Antigen Detection

#### 2.5.1. Lateral Flow Immunoassay (AG Rapid Kit—p30 Detection)

To detect viral antigens released into the supernatant, the VDRG^®^ASFV Ag Rapid Kit (Median Diagnostics Inc., Chuncheon, Gangwon-do, Republic of Korea) was used. The detection limit of this kit for the p30 antigen is 11.5 ng [26]. Antigen detection was performed on inoculum-0, inoculum-6, supernatants-6 and supernatants-12 (Figure 1). All samples were processed according to the manufacturer’s protocol. The presence of a visible test band indicated a positive result. The results were determined as positive or negative, and for positive bands, band intensity was quantified using ImageJ (software version 1.54; National Institutes of Health, Bethesda, MD, USA) and recorded.

#### 2.5.2. FITC Immunofluorescence (p54 Detection) and DAPI

Six days after inoculating Vero cells with Inoculum-0, cells were fixed with 80% acetone for 10 min at 4 °C and washed three times with PBS. Cells were blocked with 5% fetal bovine serum (FBS) in PBS for 30 min and incubated overnight at 4 °C with a primary antibody specific to ASFV p54 (1:200 dilution in blocking buffer, GeneTex, Irvine, CA, USA). After washing, the cells were incubated with FITC-conjugated goat anti-mouse IgG (whole molecule) secondary antibody (Sigma-Aldrich, St. Louis, MO, USA) at a 1:400 dilution for 1 h at room temperature. DAPI (Invitrogen, Carlsbad, CA, USA) was used to counterstain nuclei. Fluorescence was visualized using a fluorescence microscope (OLYMPUS, IX73, Tokyo, Japan), and p54 expression was qualitatively evaluated.

### 2.6. Cytotoxicity and Viability Assays

#### 2.6.1. LDH Release Assay

Cytotoxicity was quantified by measuring lactate dehydrogenase (LDH) release into the culture supernatant, reflecting cellular membrane integrity. At each time point, starting every 24 h after cell inoculation, 50 µL of supernatant was collected from each well and assayed using the Cytotox96^®^Non-Radioactive Cytotoxicity Assay kit (Promega Corporation, Madison, WI, USA) following the manufacturer’s protocol. Absorbance was measured at 490 nm using a microplate reader (Multiskan SkyHigh; Thermo Fisher Scientific, Waltham, MA, USA), and the results were normalized against maximum LDH release, determined by treating control wells with 10× Lysis Solution (included in kit) for complete lysis.

#### 2.6.2. MTT Assay

After collecting supernatants for LDH analysis, cell viability was assessed using the MTT assay. Wells were incubated with 20 µL of MTT solution (5 mg/mL in DMEM) prepared using the MTT Cell Proliferation Assay Kit (Abcam plc, Cambridge, UK) for 3 h at 37 °C. Following incubation, the medium was removed, and 150 µL of the MTT solvent was added to solubilize the formazan crystals. Absorbance was measured at 590 nm, and viability was expressed as a percentage of untreated control cells.

### 2.7. Statistical Analysis

Each experiment was performed in triplicate, and the data were expressed as mean ± standard deviation (SD). Statistical analysis was performed using GraphPad Prism (v8.4.3; GraphPad Software, San Diego, CA, USA). Statistical significance was evaluated by a one-way analysis of variance (ANOVA). A *p*-value of <0.05 was considered statistically significant.

## 3. Results

### 3.1. Cell Damage over Time

‘Inoculum-0’, prepared through the disinfectant efficacy testing process, was inoculated into Vero cells, and cytotoxic effects from disinfectants, as well as ASFV-induced CPEs, were observed daily for six days using a scoring system (Figure 2). In the A group, cells exhibited typical ASFV-induced CPEs, characterized by rounding and detachment, which progressed gradually. By DPI-6, approximately 80% of the cells had detached, though some remained adhered to the plate surface. In the D groups, 175× and 275× caused severe cytotoxicity starting from DPI-1, leading to immediate cell detachment and floating. The affected cells exhibited an irregular and distorted morphology distinct from ASFV-induced CPE. The 375× dilution resulted in slower cytotoxic progression; however, by DPI-6, nearly all cells had detached. In the AD groups, 175× and 275× displayed cytotoxic patterns similar to those in the D groups. However, in the 375× condition, cytotoxicity was notably reduced, resembling an ASFV-induced CPE instead. By DPI-6, approximately 50% of the cells remained attached, suggesting a mitigated cytotoxic effect compared to lower dilutions.

### 3.2. Antigen Detection Using AG Rapid Kit over Time

The first p30 antigen detection assay was performed using ‘Inoculum-0’, which was prepared according to the APQA disinfectant efficacy test protocol. In the A group, a faint positive band was detected, whereas all the AD groups exhibited positive bands across all dilutions. After six days of incubation (‘Inoculum-6’), the A group maintained the same detection pattern, while the AD groups showed altered band intensities (initial: 175× (73.2%) > 275× (55.1%) > 375× (43.9%); post incubation: 375× (47.1%) > 275× (45.7%) > 175× (43.1%)) (Figure 3B). Subsequent inoculation into Vero cells resulted in ‘Supernatants-12’, where the A group exhibited strong positive band. In contrast, the AD groups showed only a faint band at 375×, indicating reduced antigen detection over time. Similarly, in ‘Supernatants-6’ (immediate inoculation), the A group showed a stronger p30 antigen signal, whereas disinfectant-treated samples exhibited reduced antigen detection, particularly at 175×. Overall, the AD groups initially showed higher antigen detection levels but exhibited a time-dependent decrease, suggesting a progressive reduction in antigen presence (Figure 3A).

### 3.3. Detection of ASFV Antigen (FITC) and Nuclei Integrity (DAPI) at DPI-6

As shown in Figure 4, strong green FITC fluorescence signals were clearly observed in the A group, indicating active p54 antigen expression. Corresponding bright field images revealed distinct cytopathic effects, such as cell rounding and detachment. DAPI staining showed that most nuclei remained intact, and the merged image further supported the presence of viable cells undergoing gradual CPE progression. In contrast, no FITC signals were detected in any of the AD groups (175×, 275×, and 375×), suggesting the effective inhibition of ASFV replication. Notably, in the 175× and 275× groups, DAPI staining showed a marked reduction in intact nuclei, consistent with severe cytotoxicity. However, in the 375× group, more nuclei were preserved, indicating reduced cytotoxic effects at higher dilution. The negative control (N.C.) group exhibited no FITC signal and maintained normal cell morphology, with strong DAPI staining and no signs of CPE, validating the specificity of the observed effects.

### 3.4. Cytotoxicity Analysis Based on LDH Release and MTT Assay

At DPI-1, LDH release varied depending on disinfectant concentration and the presence of ASFV. High LDH levels were observed in the D 175×, AD 175×, and AD 275× groups. Moderate LDH release was detected in the A and the D 275× groups, while the D 375× and AD 375× groups showed low LDH levels. By DPI-4, LDH levels increased across all groups, showing a trend toward convergence. Notably, the A group, which exhibited gradual CPE progression, showed a sharp increase (88%) in LDH release between DPI-4 and DPI-5 (Figure 5A). Cumulative cytotoxicity analysis based on LDH release showed similarly high toxicity in the AD 175×, AD 275×, and A groups, while the AD 375× group exhibited the lowest cytotoxicity (Figure 5C). MTT assay results at DPI-6 further supported the LDH findings. The AD 375× group showed the lowest cytotoxicity, with fewer than 40% of cells affected and a relatively high cell viability (56.9%), consistent with the low LDH levels. In contrast, the AD 275× and AD 175× groups exhibited over 80% cytotoxicity, comparable to their respective the D groups. The A group showed a viability of approximately 36%, higher than all groups (13–15%) except for AD 375× (Figure 5B,D).

To evaluate cytotoxic stability, the coefficient of variation (CV) of daily LDH release was calculated. In this study, CV reflected temporal fluctuations in LDH levels rather than variability between replicates (Table 1). The D 175× group showed strong and consistent cytotoxicity with a mean LDH release of 44.6% and a CV of 24.9%, indicating relatively stable LDH dynamics. In contrast, the D 375× group exhibited lower LDH release (32.9%) but a higher CV (74.6%), suggesting overall low cytotoxicity but greater temporal variability. The A group showed moderate LDH release (50.2%) with a CV of 52.7%, indicating continuous virus-induced CPE with some variability over time. Among the AD groups, the AD 375× condition showed the lowest LDH release (29.4%) but the highest CV (92.99%), suggesting minimal overall cytotoxicity but irregular cell damage over time. In contrast, the AD 175× group showed the highest LDH release (51.5%) and the lowest CV (24%), indicating rapid and consistent cytotoxicity with minimal daily variation (Table 1). These CV values reflect the temporal dynamics of cytotoxicity; lower CVs (e.g., AD 175×) indicate steady and sustained toxicity, while higher CVs (e.g., AD 375×) suggest variable, time-dependent effects.

To synthesize the findings from all experimental groups, including virological markers and cell viability assays, a comprehensive summary table was generated (Table 2). This table consolidates key outcomes such as damage scores, antigen detection (the rapid kit and FITC), LDH release, and MTT viability.

## 4. Discussion

The primary objective of this study was to establish a cell-based analytical framework capable of clearly distinguishing virus-induced cytopathic effects (CPEs) from disinfectant-induced cytotoxicity during the evaluation of ASFV disinfectant efficacy. This framework is intended to contribute to more accurate assessments in future disinfectant authorization processes and disease control strategies.

Although a variety of chemical disinfectants—such as oxidizing agents, acids, aldehydes, and quaternary ammonium compounds—are widely used for the control of enveloped viruses including ASFV, there remains a notable lack of standardized and laboratory-based protocols for evaluating their virucidal efficacy. Conventional methods such as virus titration, antigen detection, CPE observation, and metabolic or biochemical assays (e.g., MTT, LDH) each have their strengths and inherent limitations. These limitations become particularly problematic under conditions where viral infection and chemical toxicity coexist, making the interpretation of results more complex [12,14,18,38].

During the course of this study, two major factors contributing to interpretative ambiguity were identified. First, under a 375-fold dilution of Virkon^®^ S, cells co-treated with ASFV and disinfectant showed higher viability and lower cytotoxicity than those treated with disinfectant alone. This finding suggests that ASFV infection may confer a transient cytoprotective effect, potentially through the virus-mediated modulation of host responses [39]. Second, p30 antigen signals were unexpectedly stronger in the virus + disinfectant group than in the untreated group, based on a rapid detection kit targeting p30. This likely reflects envelope disruption by the disinfectant, exposing internal proteins like p30, which is involved in viral entry and targeted by diagnostics and neutralizing antibodies [40,41]. Virkon^®^ S is known to disrupt the ASFV lipid envelope [24], resulting in antigen exposure [42]. Similar findings were reported previously using EM and immunogold labeling, which revealed damage to the capsid and exposure of internal antigens such as p17 and p72 [43]. As p30 and p54 are also inner envelope proteins, their increased accessibility is plausible following membrane disruption.

Beyond identifying these sources of interpretative ambiguity, a particularly noteworthy finding of this study is that effective ASFV inactivation was achieved even at a 375-fold dilution—lower than the manufacturer’s recommended 275-fold dilution—while simultaneously exhibiting reduced cytotoxicity. This result indicates that a more precise determination of working dilutions, tailored to actual use conditions, may be feasible. It also provides a scientific rationale for revisiting recommended concentrations based on quantitative cell-based data. Such an approach could contribute to more reliable efficacy testing in regulatory reviews while also helping to prevent the excessive or inappropriate use of disinfectants.

Based on these findings, we propose a multi-parametric evaluation strategy that integrates morphological observation, antigen detection, and LDH/MTT-based cytotoxicity assays (Figure 6). In particular, IFA targeting virus-specific antigens serve as a key tool to accurately determine viral inactivation, even under conditions with ambiguous CPEs or mixed cytotoxicity. While this study employed ASFV p54 as a target antigen, the same approach is readily extendable to other animal viruses using their respective diagnostic or immunogenic antigens—such as the nucleocapsid (N) protein of PEDV, the viral protein 1 (VP1) of FMDV, or the nucleoprotein (NP) of AIV.

This study has certain limitations. First, experiments were conducted only in Vero cells using a genotype I ASFV strain. However, as explained in the Introduction, the purpose was not to compare viral replication across cell types but to establish a practical framework for cytotoxicity–CPE distinction. Vero cells were chosen for their reproducibility and scalability. Validation in PAMs or wild boar lung (WSL) cells may further support the applicability of this method. Second, while only Virkon^®^ S was tested, its selection as a representative disinfectant and the rationale for the dilution range were clearly described. Future studies should expand this framework to other disinfectant classes and testing conditions to assess its broader generalizability. Lastly, although individual elements of this approach have been applied in studies of other enveloped viruses, an integrated strategy to distinguish virus-induced CPEs from disinfectant-induced cytotoxicity has not been widely reported. This study may thus serve as a reference for future application to diverse virus–disinfectant systems.

## 5. Conclusions

This study highlights the importance of clearly distinguishing disinfectant-induced cytotoxicity from virus-induced CPE in ASFV disinfectant efficacy testing. We established a multi-step analytical framework integrating morphological observation, antigen detection, and cell viability assays, which enabled a more accurate and reproducible interpretation of results. Although developed using a Vero cell-adapted ASFV strain and a single disinfectant, this framework can be adapted to other enveloped viruses and disinfectants by adjusting antigen targets and cytotoxicity thresholds. It offers a practical and scalable tool for improving the reliability and standardization of laboratory-based disinfectant efficacy testing.

## Figures and Tables

**Figure 1 pathogens-14-00451-f001:**
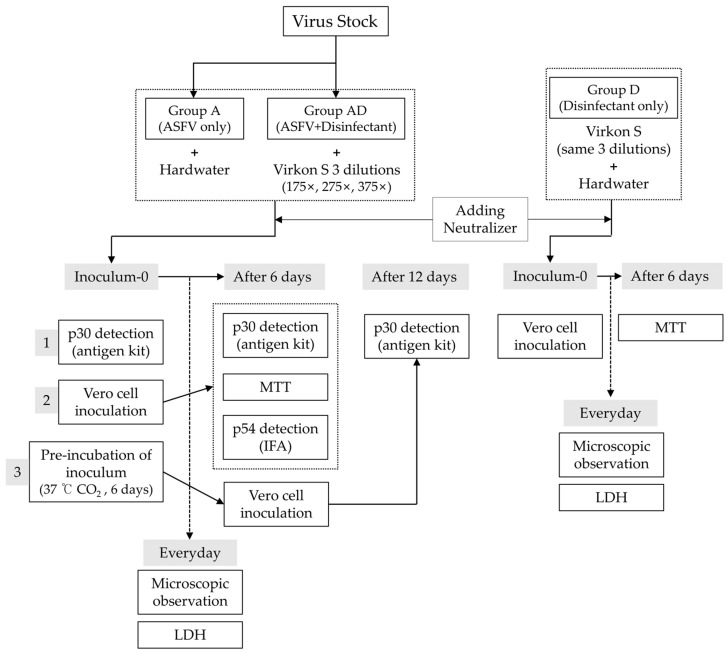
An overview of the experimental design.

**Figure 2 pathogens-14-00451-f002:**
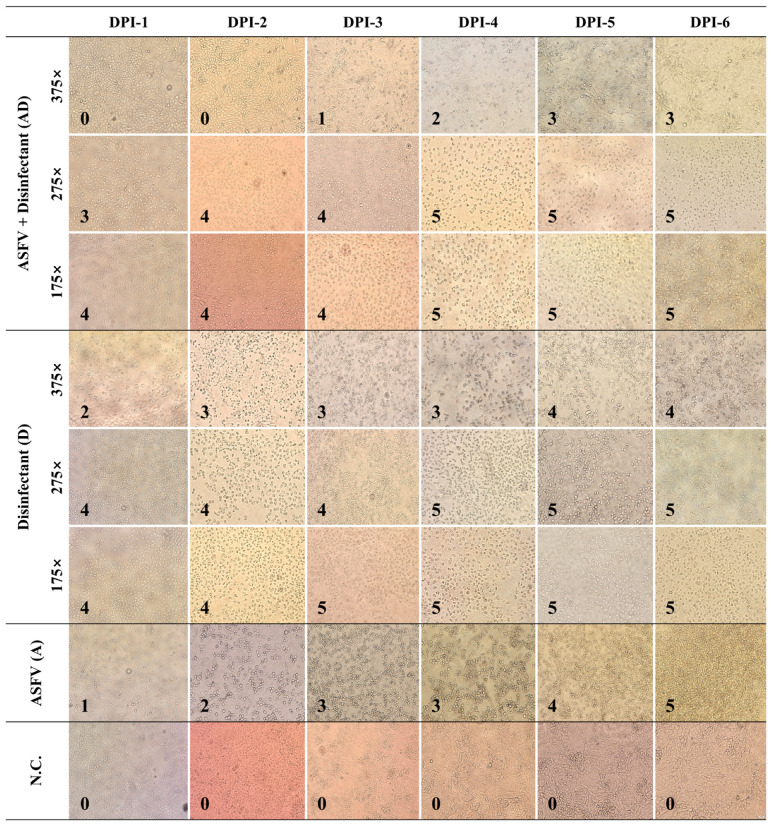
Damage scores in Vero cells exposed to ASFV and disinfectant treatments. Bright-field images of Vero cells were captured following treatment with ASFV alone (A), disinfectant only (D), or ASFV plus disinfectant (AD) at various dilutions (175×, 275×, 375×) and observed daily from 1 to 6 days post inoculation (DPI). Images were acquired at a total magnification of 200× using an Olympus IX73 inverted microscope equipped with a 20×/0.40 objective. All images are shown at the same scale for comparative purposes; therefore, individual scale bars are not included. Scoring system for cell damage: 0: No visible damage; cells appear healthy; 1: Minimal damage; slight rounding or detachment of a few cells; 2: Mild damage; increased rounding and detachment of cells (~10–20%); 3: Moderate damage; significant rounding, detachment, and cell loss (~30–50%); 4: Severe damage; widespread cell rounding and detachment (~50–80%); 5: Complete damage; almost all cells are detached or destroyed (>80%).

**Figure 3 pathogens-14-00451-f003:**
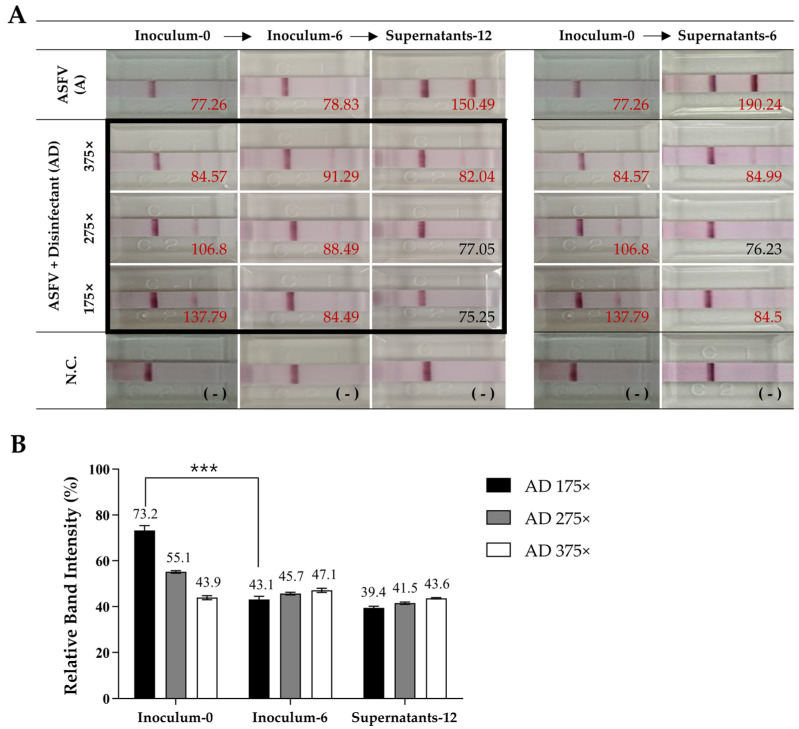
Antigen detection results for ASFV with different disinfectant treatments. (**A**) Lateral flow assay (LFA) results for ASFV detection after treatment with various disinfectants. Band intensities were quantified using ImageJ, and values are shown next to each band. Positive results are highlighted in red, and negative results are indicated in black. (**B**) The quantification of band intensities from (**A**), focusing on the groups outlined in black. The strongest signal, “Supernatants-6” from the ASFV-only group (190.24), was set as 100%, and all other values were normalized accordingly. Asterisks indicate statistically significant differences between groups (*** *p* < 0.001), determined by a one-way ANOVA.

**Figure 4 pathogens-14-00451-f004:**
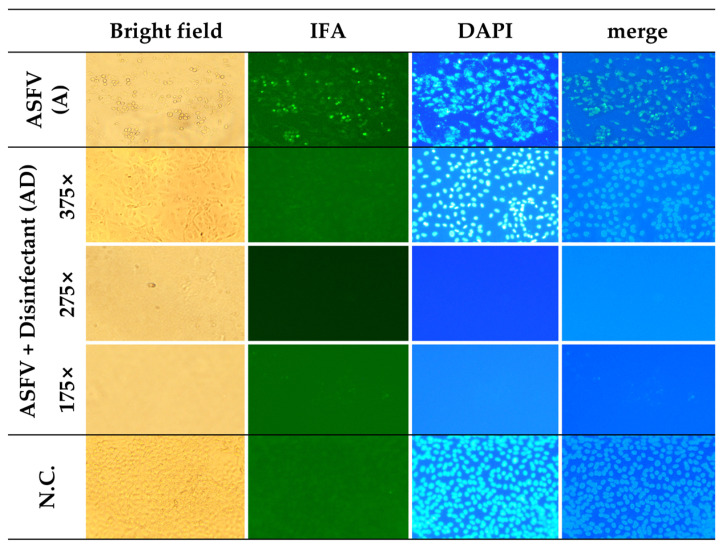
Detection of ASFV antigen (p54) and cellular integrity under different treatment conditions. Vero cells were treated with ASFV alone, ASFV combined with disinfectants at dilution rates of 175×, 275×, and 375×, or left untreated (N.C.). Cells were examined by bright-field microscopy, an indirect immunofluorescence assay (IFA; anti-p54, green), and DAPI staining (blue). The ASFV antigen was detected only in the ASFV-only group. Nuclear damage increased with higher disinfectant concentrations, while the 375× dilution group showed improved nuclear preservation. All images were acquired at the same magnification using an Olympus IX73 inverted microscope equipped with a 20×/0.40 Ph1 objective (total magnification: 200×). Images were cropped for layout consistency.

**Figure 5 pathogens-14-00451-f005:**
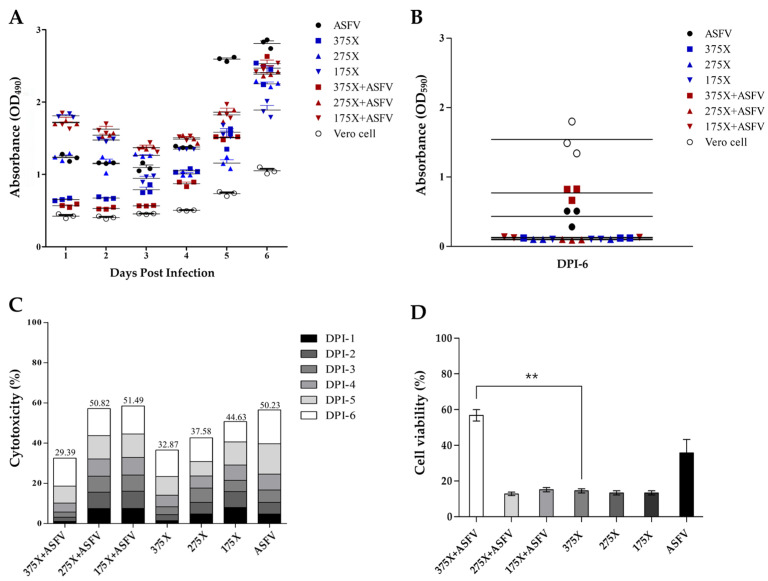
LDH release and MTT assay-based cytotoxicity analysis across experimental groups. (**A**) Daily LDH release (%) measured over six days (DPI 1–6) to assess cytotoxicity. (**B**) MTT assay absorbance values on DPI-6 to evaluate cell viability. (**C**) Cumulative LDH-based cytotoxicity (%) across the experimental period. (**D**) MTT assay-based cell viability (%) on DPI-6. Statistically significant differences were observed between the 375× + ASFV group and the D 375× group (*p* < 0.01), as determined by a one-way ANOVA followed by post hoc analysis. Asterisks (**) indicate *p* < 0.01.

**Figure 6 pathogens-14-00451-f006:**
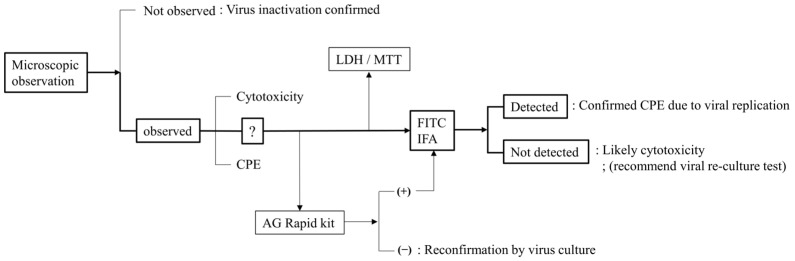
A systematic workflow to differentiate CPEs and cytotoxicity.

**Table 1 pathogens-14-00451-t001:** Cytotoxicity assessment based on LDH release.

Group		Mean (%) ± SD for Each Day							Total		
		DPI-1	DPI-2	DPI-3	DPI-4	DPI-5	DPI-6		Mean (%)	SD	CV (%)
**ASFV (A)**		28.8 ± 1.7	30.3 ± 0.2	31 ± 1.3	41.9 ± 2.1	83.2 ± 6.6	86.3 ± 2		50.2	26.5	52.7
**Disinfectant** **(D)**	**375×**	9.3 ± 0.5	16.7 ± 0.4	20.2 ± 1.5	30 ± 2.0	50.1 ± 5.4	70.3 ± 6.0		32.9	24.5	74.6
	**275×**	29.1 ± 1.4	30 ± 3.3	37.2 ± 1.2	29.1 ± 2.2	37.2 ± 4.6	64.9 ± 1.5		37.6	12.4	33.1
	**175×**	48.3 ± 0.7	39.7 ± 0.7	26.1 ± 1.3	40.4 ± 0.9	61.7 ± 12.2	49.6 ± 4.5		44.6	11.1	24.9
**ASFV + Disinfectant** **(AD)**	**375×**	6.4 ± 0.6	11.8 ± 0.4	13.6 ± 0.1	24.3 ± 0.3	46.6 ± 0.9	75.8 ± 3.1		29.4	27.3	92.9
	**275×**	45.1 ± 0.6	41.0 ± 0.4	40.1 ± 0.6	43.8 ± 0.8	61.3 ± 3.1	69.7 ± 1.5		50.8	12.1	23.7
	**175×**	45.4 ± 3.1	44.1 ± 1.9	39.9 ± 2.1	45.0 ± 1.8	61.6 ± 2.3	72.8 ± 2.1		51.5	12.4	24.0

Mean: Average cytotoxicity percentage measured over six days (DPI-1 to DPI-6). SD: standard deviation of the cytotoxicity measurements, showing the dispersion of values around the mean. CV (%): a measure of relative variability; a higher CV indicates greater variability in the data, while a lower CV indicates more consistent results.

**Table 2 pathogens-14-00451-t002:** Summary of experimental outcomes for ASFV and disinfectant treatments.

		Damage Scores (DPI 1~6)	Rapid Kit (p30), %(DPI 0, 6, 12)	FITC (p54)	LDH Cytotoxicity, % (DPI 1~6)	MTT Viability, % (DPI 6)
**ASFV (A)**		1/2/3/3/4/5	40.5/41.4/79	Pos	28.8/30.3/31.0/41.9/83.2/86.3	35.9
**ASFV + Disinfectant** **(AD)**	**175×**	4/4/4/5/5/5	73.2/43.1/39.4	Neg	45.4/44.1/40.0/45.0/61.6/72.8	15.2
	**275×**	3/4/4/5/5/5	55.1/45.7/41.5	Neg	45.1/41.0/40.1/43.8/61.3/69.7	12.9
	**375×**	0/0/1/2/3/3	43.9/47.1/43.6	Neg	6.4/11.8/13.6/24.3/46.6/75.8	56.9
**Disinfectant** **(D)**	**175×**	4/4/5/5/5/5	-	-	48.3/39.7/26.1/40.4/61.7/49.6	13.4
	**275×**	4/4/4/5/5/5	-	-	29.1/30.0/37.2/29.2/37.2/64.9	13.4
	**375×**	2/3/3/3/4/4	-	-	9.3/16.7/20.2/30.7/50.1/70.3	14.6

## Data Availability

The data presented in this study are available in the article.

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
