# Peer review of "A Practical Framework for ASFV Disinfectant Evaluation: Differentiating Cytopathic Effects from Cytotoxicity via Integrated Analytical Methods"

_pathogens, 2025, doi:10.3390/pathogens14050451_

Round 1
Reviewer 1 Report
Comments and Suggestions for Authors
This manuscript addresses a critical issue in the field of ASFV disinfectant efficacy testing. This method substantially and systematically improves the accuracy and reproducibility of in vitro assessments. The topic is cutting-edge and the experimental design is reasonable. The following comments and questions are provided for further improvement:
1. Please provide more detailed information regarding the antibodies used and other details are essential for improving the reproducibility of the study.
2. It is recommended to briefly discuss the feasibility and preliminary results of applying this method to a range of disinfectants, different viruses, or alternative host cell systems, so as to clarify the generalizability and limitations of the approach. Please consider supplementing the manuscript with extra literature review on similar analytical approaches used for other enveloped viruses as the authors mentioned.
3. The experiments in this study utilize only Vero cell lines and a single adapted ASFV genotype I strain. However, ASFV exhibits significant differences in replication and CPE in PAMs, ex vivo infection scenarios. Please clarify whether the applicability of this method has been evaluated in PAMs or other relevant cell types like WSL cells. Given that only one commercial disinfectant at limited dilutions was tested, it is strongly recommended to discuss the generalizability and limitations of the protocol with respect to other major disinfectant classes and under varying physicochemical and biological conditions (such as different chemical compositions, pH, and organic load).
Author Response
Comment 1: Please provide more detailed information regarding the antibodies used and other details are essential for improving the reproducibility of the study.
Response 1: We appreciate the reviewer’s suggestion. To enhance the reproducibility of our study, we have added detailed information about the antibodies used, including clone names, manufacturers, and working dilutions, in the Materials and Methods section (page 6, lines 205-207). This additional information will enable readers to replicate the procedures more precisely.
Comment 2: It is recommended to briefly discuss the feasibility and preliminary results of applying this method to a range of disinfectants, different viruses, or alternative host cell systems, so as to clarify the generalizability and limitations of the approach. Please consider supplementing the manuscript with extra literature review on similar analytical approaches used for other enveloped viruses as the authors mentioned.
Response 2: We appreciate the reviewer’s insightful comment regarding the generalizability of our approach to other disinfectants, viruses, and host cell systems. In response, we have addressed this point in the revised manuscript by explicitly discussing these limitations and the potential for broader application in the Discussion section (page 13, lines 416–420).
Comment 3: The experiments in this study utilize only Vero cell lines and a single adapted ASFV genotype I strain. However, ASFV exhibits significant differences in replication and CPE in PAMs, ex vivo infection scenarios. Please clarify whether the applicability of this method has been evaluated in PAMs or other relevant cell types like WSL cells. Given that only one commercial disinfectant at limited dilutions was tested, it is strongly recommended to discuss the generalizability and limitations of the protocol with respect to other major disinfectant classes and under varying physicochemical and biological conditions (such as different chemical compositions, pH, and organic load).
Response 3: We thank the reviewer for the valuable comment regarding the use of a single cell line, a single ASFV genotype, and a single disinfectant, as well as the generalizability of the proposed method. These points have already been clearly addressed in the manuscript. Specifically, in the Introduction (pages 2–3, lines 88–105), we explain the rationale for using a Vero cell-adapted ASFV genotype I strain and Virkon® S as a representative disinfectant. Vero cells were selected due to their scalability, reproducibility, and common use in ASFV research, making them suitable for establishing a standardized analytical framework. Likewise, Virkon® S was chosen based on its international use and regulatory recognition, serving as an appropriate model disinfectant to validate the proposed method. The goal of this study was not to compare virus behavior across multiple systems, but to establish and validate a reliable method for differentiating virus-induced CPE from disinfectant-induced cytotoxicity.
In addition, we acknowledge the limitations of this approach in the Discussion section (page 13, lines 408–416). We explicitly state that the study was limited to one cell type and one disinfectant and recommend future studies to apply the framework to other ASFV-relevant cell lines such as PAMs or WSL cells. We also note that the method should be tested against a broader range of disinfectant classes (e.g., quaternary ammonium compounds, oxidizing agents, organic acids) and under varying physicochemical conditions such as different pH levels, chemical compositions, and organic loads.
We hope this clarification demonstrates that the manuscript appropriately addresses the reviewer’s concerns and presents a balanced discussion of the scope and limitations of the study.
Reviewer 2 Report
Comments and Suggestions for Authors
Dear Authors,
The study summary is interesting, but its detail and scope do not allow to the reader to fully reflect the actual text and do not correspond properly with the text of the work.
I suggest verification of the summary and/or possibly adjusting the title of the manuscript to the content of the main body of the text.
The introduction should be expanded to include a historical part regarding ASFV (main global spreads), especially since this aspect and problems with combating to this virus constitute the essence of the work.
The analytical methods have been described correctly, but a great deal of uncertainty is caused by the scheme and the adopted operational protocol in the scope of the experimental methodology, which requires explanation. The essence of the experiment and the selected dilutions have not been clearly explained. It is not understandable why only Virkon has a reference in the experiment, despite the presence of other analogous commercial products on the global market.
Contrary to appearances and description, Fig. 1. does not illustrate an experimental procedure, but this kind of full design would be highly appreciated, including different dilutions which were used in the study.
In the reviewer's opinion, the text of the article must be definitely refined, with a clear and specific research goal that will fully correspond to the obtained results. Currently, the reader may get the impression that the presented results - although they are interesting and probably have practical significance - are an introduction to another study, as literature ground.
In summary: the current version of the article needs to be corrected. Than the reader will receive a clear and unambiguous information from this text. Current version of the manuscript does not meet such a criterias what was presented above.
Kind regards!
Author Response
- Comment 1: I suggest verification of the summary and/or possibly adjusting the title of the manuscript to the content of the main body of the text.
Response 1: Thank you for your comment. In response, we have carefully revised the title of the manuscript to better reflect the main focus and content of the study. Additionally, the Abstract has been completely rewritten to clearly summarize the research objectives, methodology, key findings, and significance. We believe these changes improve the clarity and consistency between the abstract, title, and main text of the manuscript.
- Comment 2: The introduction should be expanded to include a historical part regarding ASFV (main global spreads), especially since this aspect and problems with combating to this virus constitute the essence of the work.
Response 2: We appreciate the reviewer’s helpful suggestion. In response, we have expanded the Introduction section to include a historical overview of the global spread of African swine fever virus (ASFV), highlighting major outbreaks and the ongoing challenges in controlling the disease. This information has been added on pages 1–2, lines 37–47 of the revised manuscript to better contextualize the significance of the study.
- Comment 3: The analytical methods have been described correctly, but a great deal of uncertainty is caused by the scheme and the adopted operational protocol in the scope of the experimental methodology, which requires explanation.
Response 3: We thank the reviewer for pointing out the need for improved clarity regarding the experimental scheme and operational protocol. In response, we have completely restructured Figure 1 to present the workflow more clearly and logically. The figure legend has also been revised to provide a more detailed explanation of each step in the analytical process. We believe these changes significantly improve the readability and transparency of the experimental methodology.
- Comment 4: The essence of the experiment and the selected dilutions have not been clearly explained. It is not understandable why only Virkon has a reference in the experiment, despite the presence of other analogous commercial products on the global market.
Response 4: We thank the reviewer for the valuable comment. To clarify the rationale for using Virkon® S and the selected dilutions, we have revised the Introduction (pages 2–3, lines 88–96) to explicitly explain the disinfectant selection and experimental design.
- Comment 5: In the reviewer's opinion, the text of the article must be definitely refined, with a clear and specific research goal that will fully correspond to the obtained results.
Response 5: We thank the reviewer for this constructive comment. In response, we have thoroughly revised the description of the experimental procedure in the Materials and Methods section (page 4, lines 153–178) to clearly outline the full study design, including all tested dilutions and treatment groups. Additionally, Figure 1 has been updated to more accurately illustrate the experimental workflow and to reflect the full range of conditions evaluated in the study.
- Comment 6: In the reviewer's opinion, the text of the article must be definitely refined, with a clear and specific research goal that will fully correspond to the obtained results. Currently, the reader may get the impression that the presented results - although they are interesting and probably have practical significance - are an introduction to another study, as literature ground.
Response 6: We appreciate the reviewer’s thoughtful feedback. In response, we have revised the title, abstract, discussion, and conclusion sections to clearly articulate the research objective and to ensure stronger alignment between the aim of the study and the presented results. These revisions aim to emphasize that the current findings represent a complete and practical framework, rather than a preliminary step toward future work.
Reviewer 3 Report
Comments and Suggestions for Authors
In the manuscript, the authors describe the development of a method for testing the effectiveness of disinfectants that allow to differentiate between CPE caused by viral activity or the action of a disinfectant. The authors touch upon an important issue, since it is often unclear what causes CPE when testing desinfectants on cell cultures. The work demonstrates that a combination of antigen detection methods with cell viability determination methods allows to identify the cause of CPE. Thus, the authors examined an important problem and is relevant at the present time. The manuscript provides a detailed description of the methods used, which allows one to reproduce the work done. The photographs of the results confirm the presented results. The description and analysis of the data obtained is detailed and correct. The English language is understandable and does not require additional correction.
The disadvantages of the work include, probably, the lack of universality of the method. To reproduce the method on another virus, it is necessary to study its biology and the timing of antigen expression. However, in the case of the African swine fever virus, the method is well developed.
Minor shortcomings include:
- The captions to the photographs of the cell cultures do not indicate the degree of magnification of the microscope.
-Typo detected in line 189 (Olympus).
Author Response
- Comment 1: The disadvantages of the work include, probably, the lack of universality of the method. To reproduce the method on another virus, it is necessary to study its biology and the timing of antigen expression. However, in the case of the African swine fever virus, the method is well developed.
Response 1: We thank the reviewer for this insightful comment. We agree that the generalizability of the proposed method may be limited by virus-specific biological characteristics, including antigen expression kinetics. To address this point, we have included a corresponding discussion in the limitations section of the Discussion (pages 12–13, lines 408–420), noting that adaptation of the framework to other viruses would require prior knowledge of viral biology and marker selection. Nonetheless, we emphasize that the method is well optimized for ASFV, as acknowledged by the reviewer.
- Comment 1: The captions to the photographs of the cell cultures do not indicate the degree of magnification of the microscope.
Response 1: We appreciate the reviewer’s observation. In response, we have updated the figure legends for Figures 2 and 4 to include the magnification settings used during image acquisition. This information will help readers better interpret the scale and cellular morphology presented in the images.
- Comment 1: Typo detected in line 189 (Olympus).
Response 1: We appreciate the reviewer’s attention to detail. The typo has been corrected in line 209 where “Olypus” was revised to “Olympus” in the updated manuscript.
Round 2
Reviewer 2 Report
Comments and Suggestions for Authors
The manuscript was revised and changed by the Authors after first review.
Present version of the article is acceptable and now I may recommend mentioned text to be publish in Pathogens.
Gratulations for the Authors!